# Regulatory T Cell-Enhancing Therapies to Treat Atherosclerosis

**DOI:** 10.3390/cells10040723

**Published:** 2021-03-24

**Authors:** Hafid Ait-Oufella, Jean-Rémi Lavillegrand, Alain Tedgui

**Affiliations:** 1Paris Cardiovascular Research Center—PARCC, Université de Paris, INSERM UMR-S 970, 75012 Paris, France; jrlavillegrand@gmail.com (J.-R.L.); alain.tedgui@inserm.fr (A.T.); 2AP-HP (Assistance Publique-Hôpitaux de Paris), Hôpital Saint-Antoine, Sorbonne Université, 75012 Paris, France

**Keywords:** atherosclerosis, immunity, cytokines, T lymphocytes, regulatory T cells

## Abstract

Experimental studies have provided strong evidence that chronic inflammation triggered by the sub-endothelial accumulation of cholesterol-rich lipoproteins in arteries is essential in the initiation and progression of atherosclerosis. Recent clinical trials highlighting the efficacy of anti-inflammatory therapies in coronary patients have confirmed that this is also true in humans Monocytes/macrophages are central cells in the atherosclerotic process, but adaptive immunity, through B and T lymphocytes, as well as dendritic cells, also modulates the progression of the disease. Analysis of the role of different T cell subpopulations in murine models of atherosclerosis identified effector Th1 cells as proatherogenic, whereas regulatory T cells (Tregs) have been shown to protect against atherosclerosis. For these reasons, better understanding of how Tregs influence the atherosclerotic process is believed to provide novel Treg-targeted therapies to combat atherosclerosis. This review article summarizes current knowledge about the role of Tregs in atherosclerosis and discusses ways to enhance their function as novel immunomodulatory therapeutic approaches against cardiovascular disease.

## 1. Introduction

Atherosclerosis is the result of an inflammatory response triggered by the sub-endothelial accumulation of low-density lipoprotein (LDL) cholesterol and its subsequent oxidation in arteries. Until recently, this view was mainly based on experimental evidence or clinical association studies. Yet recent clinical trials highlighting the clinical benefit of inflammation-targeted therapy with anti-interleukin (IL)-1β neutralizing antibodies [1] or colchicine [2,3] for the prevention of recurrent events in patients recovering from an acute coronary syndrome (ACS), or in the stable phase of coronary artery disease, have provided a direct proof of concept that atherosclerosis is a chronic inflammatory disease, as claimed by R. Ross at the end of the last century [4]. Although monocytes/macrophages constitute the central cells responsible for the innate immune response and are mandatory for the formation of foam cells, adaptive immune responses, through T and B cells, as well as dendritic cells (DCs), have been shown to substantially regulate atherosclerosis in experimental models [5]. Conventional B2 cells have been shown to exert proatherogenic activities, whereas natural B1 cells producing anti-oxidized LDL (oxLDL) IgM display antiatherogenic properties [6]. Animal studies have implicated cluster of differentiation (CD)4^+^ T helper type 1 (Th1) cells as important contributors to the initiation of atherosclerosis, whereas the role of Th2 and Th17 cells remains debated [5]. Importantly, regulatory T cells (Tregs) expressing the IL-2 receptor α-chain (CD25) and the transcription factor FoxP3 have been shown to protect against atherosclerosis [7]. Our understanding that the imbalance between the suppressive activity of Tregs and activation of harmful Th1 responses contributes to atherosclerosis development has opened new opportunities for enhancing immune tolerance by stimulating Tregs in order to promote atheroprotective immunity. In this review, we discuss recent advances in our understanding of the roles of Tregs in atherosclerosis, and discusses ways to stimulate Treg-based immunomodulatory approaches as novel therapeutic strategies against the disease.

## 2. Role of T Cell Activation in Atherosclerosis

The presence of CD3^+^ T cells in both mouse [8] and human atherosclerotic plaques [9], and the pronounced expression of major histocompatibility complex (MHC) class II (HLA-DR) in human plaques [10] were the first evidence supporting a role for adaptive immunity in atherosclerosis. More recently, single-cell RNA sequencing (scRNA-seq) analysis confirmed the presence of both CD4^+^ and CD8^+^ T cells in mouse [11,12] and human [13] plaques. T lymphocytes are one of the earliest cells recruited in the atherosclerotic plaque [14]. More direct and convincing evidence for a role of adaptive immunity in atherosclerosis has been provided using experimental mouse models of atherosclerosis, especially *Apolipoprotein E* (*Apoe*)^−/−^ or *low-density lipoprotein receptor* (*Ldlr*)^−/−^ mice, in which human-like atherosclerotic lesions develop spontaneously or in response to a high-fat diet. Athero-prone mice crossed with T/B cell-deficient *Rag1*^−/−^ (or *Rag2*^−/−^) or *Scid/Scid* mice display reduced atherosclerosis lesions when fed a chow diet [5]. In addition, purified CD4^+^ T cells supplementation into *Scid/Scid/Apoe*^−/−^ mice reverses the beneficial effect of lymphocyte deficiency [15]. CD8^+^ T cells also contribute to atherosclerosis progression [16]. Depletion of CD8^+^ T cells with anti-CD8α or -CD8β antibodies in *Apoe*^−/−^ [17] or *Ldlr*^−/−^ mice [18] attenuated atherosclerosis by modulating monopoiesis in the bone marrow and reducing circulating monocyte levels in the blood.

The exact mechanism that orchestrates T cell immunity in atherosclerosis is complex, but accumulating evidence supports an antigen-specific-driven response. T cells derived from *Apoe*^−/−^ mice exhibit a highly restricted T cell receptor (TCR)α/β repertoire [19]. Moreover, TCRα deficiency in *Apoe*^−/−^ mice has been shown to protect against atherosclerosis [20]. In the same line, T cells from human atherosclerotic plaques respond to oxLDL in an HLA-DR-dependent manner, but not to naïve LDL [21], suggesting that oxLDL is an important antigen in atherosclerosis. In addition, the transfer of T cells pre-exposed to oxLDL into *scid/scid/Apoe*^−/−^ mice accelerates atherosclerotic lesion development [15]. However, CD4^+^ T cells recognizing epitopes derived from unmodified apolipoprotein B100 (ApoB) has also been found, first in human ApoB100 transgenic mice [22], and more recently in the blood from subjects with and without atherosclerosis, by using an MHC class II-tetramer loaded with an ApoB epitope [23]. Of note, these ApoB100-autoreactive CD4^+^ T cells are mainly comprised FoxP3+ Tregs in healthy individuals, whereas in patients with subclinical atherosclerosis, they are comprised less of Tregs and more of T-bet+ Th1 and retinoic-acid receptor-related orphan receptor (ROR)-γt+ Th17 cells. Atherosclerosis-related antigens recognized by CD8^+^ T cells have not yet been identified [16].

Other autoantigens, such as heat-shock proteins (HSPs) 60/65, have been detected in atherosclerotic plaques [24,25], and the transfer of effector T cells from *MRL-lpr* mice, which develop an autoimmune disease resembling systemic lupus erythematosus (SLE), into *Ldlr*^−/−^ mice has been shown to increase atherosclerosis [26]. SLE autoantigens are most often of nuclear origin, suggesting that antigens not specific to atherosclerosis can also be proatherogenic. Human epidemiological studies support these experimental data: the risk of cardiovascular disease (CVD) after adjustment to traditional cardiovascular risk factors is increased in patients with autoimmune disorders, including rheumatoid arthritis [27], SLE [28], or psoriasis [29]. More recently, a clinical trial conducted in cancer patients, showing higher risk for atherosclerotic cardiovascular events after treatment with immune checkpoint inhibitors, including anti-programmed death (PD)-1, anti-programmed death ligand (PDL)-1, and anti-cytotoxic T-lymphocyte antigen (CTLA)-4, that activate CD4^+^ and CD8^+^ T cells, has provided, for the first time, direct evidence in humans for a pro-atherogenic role of effector T cells in atherosclerosis [30]. Notably, a previous experimental study in *Ldlr*^−/−^ mice had reported that deficiency in PDL1/2 enhanced atherosclerosis by activating T cells [31].

## 3. Th1/Th2/Th17 Subsets

Depending on co-stimulatory or -inhibitory signaling, and pro- or anti-inflammatory cytokine production in the micro environment, naïve CD4^+^ T cells differentiate into various effector or regulatory T cell subsets upon engagement of the TCR with the antigen–MHC-II complex on an antigen-presenting cell (APC).

### 3.1. Th1 Cells

Th1 polarization is driven by inflammatory cytokines, such as interferon (IFN)-γ and IL-12, which trigger two key lineage-defining transcription factors: T-box transcription factor (TBX)-21 (also referred to as T-bet) and signal transducer and activator of transcription (STAT-4). T-bet induces the optimal production of IFN-γ and expression of the high-affinity IL-12 receptor, while inhibiting the expression of Th2 cytokines, such as IL-4 and IL-5. Th1 cells are critical for immunity against intracellular pathogens, and play a major role in the development of several autoimmune and inflammatory diseases. Two decades of animal studies have yielded evidence that atherosclerosis is a Th1-driven disease. Th1 T cells are the most abundant T-cell subset in human atherosclerotic plaques, [32], secreting IFN-γ, tumor necrosis factor (TNF)-α, and IL-2. The hallmark Th1 cytokine, IFN-γ, exerts several pro-atherogenic actions, promoting both immune and vascular (endothelial and smooth muscle) cell activation [33]. 

### 3.2. Th2 Cells

Th2 commitment is mainly induced by IL-6 and IL-13, produced by DCs, as well as OX40–OX40L interaction. Th2 cells secrete IL-4, IL-5, and IL-13. IL-4 activates STAT-6, which induces the expression of the Th2 master transcription factor, GATA-3, crucial for appropriate expression of the Th2-specific cytokines, IL-4 and IL-5. GATA-3 inhibits the production of IFN-γ and Th1 differentiation. The role of Th2 T cells in atherosclerosis remains controversial, and seems to depend on the stage of the disease, the site of the atherosclerotic lesion, the secreted Th2-specific cytokine, and the experimental model [34]. While the role of IL-4 in atherosclerosis remains unclear, other Th2-related cytokines, IL-5, IL-13, and IL-33 appear to exhibit anti-atherogenic properties. It is noteworthy that Th2 T cells share a common secretory profile with a subset of innate immune cells (ILCs), ILC-2, producing IL-5 and IL-13 but not IL-4. ILC-2 has been shown to be atheroprotective through IL-5 and IL-13 production [35].

### 3.3. Th17 Cells

Th17 commitment requires the nuclear receptors ROR-α and -γt, STAT-3, and runt-related transcription factor 1 (RUNX1). Th17 cells produce large amounts of IL-17A and to a lesser extent IL-17F, IL-22, and IL-23. In mice, TGF-β and IL-6 are required for Th17 differentiation, while IL1-β is instrumental in driving human Th17 differentiation. Next, IL-21 and IL-23 are required for Th17 proliferation and maintenance [36]. IL-6 activates the JAK/STAT-3 pathway, which is required for ROR-γt expression and function. Many experimental studies have shown that Th-specific cytokines and transcription pathways inhibit each other mutually. Th1 and Th2 cytokines, IFN-γ, and IL-4, negatively regulate Th17 differentiation, whereas IL-17 inhibits Th1 polarization, IFN-γ production, and T-bet expression [37]. Th17 cells provide protective immunity against extracellular bacteria and fungi by activating neutrophils with IL-17A and IL-17F. They also contribute to chronic inflammatory and autoimmune diseases, such as experimental autoimmune encephalomyelitis and rheumatoid arthritis [38]. Both IL-17 and IL-17-producing T cells have been detected in murine and human atherosclerotic lesions [39]. The role of IL-17 has been investigated in several mouse models of atherosclerosis, but results are controversial [39]. Interestingly, IL-17 can stimulate collagen synthesis by vascular smooth muscle cells, and increased levels of IL-17 in human carotid plaques are associated with characteristics of plaque stability [39,40]. In patients with ACS, low levels of IL-17 are associated with an increased risk of death and myocardial infarction [41]. In agreement with a pro-stabilizing role of IL-17, severe cardiovascular events have been recently reported in patients with a high cardiovascular risk after the initiation of treatment with ustekinumab (anti-IL-12/23p40 antibody), which targets the IL-17 pathway [42]. 

## 4. Treg Cells

Tregs constitute a specific subset of T cells, with several subtypes, including CD4^+^CD25^+^FoxP3^+^ Tregs, IL-10-secreting type 1 FoxP3^−^ Treg (Tr1), TGF-β-secreting type 3 Th cells (Th3), and CD8^+^ Tregs. 

### 4.1. Treg Ontogeny

To be efficient, the T cell immune response must be tightly regulated, in order to effectively respond to pathogens, prevent responses to self-antigens, and avoid mounting harmful responses to innocuous antigens. Cell-intrinsic mechanisms of tolerance in the thymus and periphery ensure the physical elimination of self-reactive T cells by clonal deletion or their functional inactivation by clonal anergy. In addition, cell-extrinsic mechanisms, mediated in part by specialized CD4^+^ T cells with immunosuppressive activity, Tregs, help regulate Th cell activity [43]. Naturally-occurring Tregs are generated in the thymus during T cell development. They comprise 5–10% of all peripheral CD4^+^ T cells. Treg lineage commitment essentially depends on TCR engagement by high avidity for self-peptide–MHC and costimulatory signals mediated by CD28, resulting in the upregulation of CD25. IL-2 or IL-15 are required for fully committed Tregs expressing the forkhead/winged helix transcription factor FoxP3, which is crucial for their development and function. Deficiency of FoxP3 in both humans and mice results in a lack of Tregs, and the development of severe systemic inflammatory diseases manifested by autoimmunity, colitis, and allergies [44]. The second route for the generation of FoxP3^+^ Tregs, termed induced Tregs (iTregs), is the differentiation of naïve CD4^+^ T cells in the periphery following TCR stimulation in the presence of IL-2 and TGF-β. Although iTregs compose only a small percentage of Tregs as a whole, this cell subpopulation is particularly enriched in certain tissues, such as the gut and maternal placenta, and crucially participates in the establishment of tolerance against commensal bacteria, foods, allergens, and the fetus in a pregnant mother [45]. In addition to the phenotypic and functional heterogeneity of Tregs, it has been suggested that Tregs can become unstable under certain inflammatory conditions and acquire a phenotype more characteristic of effector T cells, which promotes rather than suppresses inflammation [43].

### 4.2. Treg Suppressive Activity

Tregs suppress excess immune responses against a diverse range of antigens, thereby preventing autoimmune diseases and maintaining self-tolerance. They are capable of blocking effector T cells at different stages, inhibiting naïve T cell proliferation, Th1/Th2/Th17 differentiation, and T cell activation [46] through direct interaction or inhibition of APCs (Figure 1). Tregs exert suppressive activity by three different mechanisms: cell–cell contact, local production of inhibitory cytokines, and local competition for growth factors [43]. First, Tregs may kill target T cells by direct receptor–ligand interaction, releasing suppressive factors like cyclic adenosine monophosphate (cAMP) [47], combined or not with suppressive cytokines, including TGF-β [48], via gap junctions. Tregs may also limit effector T cell activity indirectly via modulating APC through reverse signaling via Treg–CTLA-4 engagement of CD80/86 on DCs [49]. The immune-suppressive actions of Tregs may be transmitted through the secretion of the anti-inflammatory cytokines IL-10 [50], TGF-β, and IL-35 [51], or induced production by APCs. In addition, expression of CD73/CD39 by Tregs promotes the local production of adenosine to downregulate immune function [52]. Treg cells can also secrete granzyme A/B and perforin during Treg–effector cell interaction [53]. Finally, Tregs may compete with effector T cells and consume cytokine signaling via receptors that contain the common γ-chain (IL-2, IL-4, and IL-7) [54].

### 4.3. Tregs and Atheroprotection

The first evidence for a role of CD4^+^CD25^+^ Tregs was provided by studies in murine models of atherosclerosis, showing that Treg depletion obtained either by gene deletion (CD80/86, CD28, inducible T-cell costimulator (ICOS)) or following treatment with CD25-neutralizing antibodies exacerbated atherosclerosis [7,55]. Thereafter, evidence from animal and human studies confirmed that decreased Treg numbers or impaired immunosuppressive Treg function promoted atherosclerosis. The precise role of FoxP3^+^ Tregs was investigated in chimeric DEREG (depletion of regulatory T cells) crossed with *Ldlr*^−/−^ mice, with specific ablation of FoxP3^+^ Tregs induced by diphtheria toxin injection. Depletion of FoxP3^+^ Tregs exacerbated atherosclerosis, but was also associated with higher plasma levels of atherogenic lipoprotein [56]. Conversely, in vivo supplementation with purified spleen CD4^+^CD25^+^ Tregs [7,57] or IL-10-producing Tr1-like Tregs [58] reduced atherosclerosis in *Apoe*^−/−^ mice and induced a more stable plaque phenotype.

CD8^+^CD25^+^ Tregs with atheroprotective functions have been found in atherosclerotic plaques of *Apoe*^−/−^ mice fed a high-fat diet [59]. Another subset of CD8^+^ Tregs that recognize self-peptides associated with Qa-1, the mouse ortholog of the human class Ib MHC HLA-E, has been identified in murine models of systemic autoimmune disease [60]. These CD8^+^ Tregs have also been shown to be potent modulators of atherosclerosis in *Apoe*^−/−^ mice through follicular helper T (Tfh) function in tertiary lymphoid organs in the aorta [61].

Analysis of the time course of Treg infiltration in atherosclerotic lesions indicates that Tregs accumulate in the aorta of *Apoe*^−/−^ mice after initiation of a high-fat diet, but their numbers decline over time under sustained hypercholesterolemia, whereas effector T cell numbers and atherosclerotic lesion size increase over the same period of time [62]. A recent study shows that under conditions of sustained hypercholesterolemia, a population of IFNγ-producing Tregs, so-called Th1/Tregs, is generated within the aorta and secondary lymphoid organs of aged *Apoe*^−/−^ mice [63], which might account for the decline in Treg numbers under a prolonged high-fat diet. A similar population of Th1/Treg expressing CCR5 has also been reported in *Apoe*^−/−^ mice on atherogenic diet for a long period of time (12–20 weeks) [64], supporting that sustained hypercholesterolemia promote plasticity in Tregs [65]. Similarly, during atherosclerosis development in western diet-fed *Apoe*^−/−^ mice, it has been shown by tracking Treg lineage that a fraction of Treg cells switches their phenotype into pro-atherogenic Tfh cells [66]. Interestingly, ApoAI treatment was able to reduce this switch by maintaining CD25 expression on Treg cells. Altogether, these experimental studies suggest that the initial CD4^+^ T cell response in atherosclerotic plaque is more oriented toward a Treg phenotype, but may switch overtime toward a pro-atherogenic effector T cell phenotype, likely due to the inflammatory and hypercholesterolemic microenvironment. Sustained hypercholesterolemia likely leads to intracellular accumulation of cholesterol and affects lipid rafts on the Treg membrane, in which CD25 is present, promoting Th1/Th17-like Treg polarization [67]. 

In human atherosclerotic plaques, Tregs are barely found during all stages of development, with less than 5% of infiltrating T cells being positive for FoxP3 [68]. Description of the immune cell landscape of advanced human atherosclerotic plaques by scRNA-seq technology has confirmed the presence of activated CD8^+^ and CD4^+^ Th1 lymphocytes, but identified only a small Treg cluster, based on the expression of FoxP3, CD25, and CTLA-4 [69]. Treg numbers have been shown to be lower in vulnerable than in stable plaques [70], but to be elevated in the thrombus attached to ruptured plaques [71]. Another level of complexity to the role of Tregs in atherosclerosis is that, in contrast to mice, FoxP3 is found in humans in several isoforms, including FoxP3Δ2 lacking exon 2, the dominant form in activated Tregs. Interestingly, the expression of FoxP3Δ2 was decreased in unstable carotid atherosclerotic plaques from patients with ischemic symptoms, compared with stable plaques from asymptomatic patients [72]. In clinical studies, reduced numbers of circulating Tregs have been reported in patients with ACS compared to patients with stable coronary artery disease (CAD) or normal coronary arteries [73,74]. Treg-suppressive functions may also be compromised in patients with ACS, with both FoxP3 and CTLA-4 expression being decreased, as well as ex vivo immune suppressive capacities [73]. Similarly, in a large prospective cohort study, increased risk for myocardial infarction was associated with low levels of circulating Tregs [75]. However, more recently, in two population-based cohorts, no association was found between CD4^+^CD25^+^CD127^−^ Tregs in peripheral blood and future myocardial infarction or stable CAD [76].

### 4.4. Mechanisms of Atheroprotection by Tregs

#### 4.4.1. Secretion of Anti-Inflammatory Cytokines

The mechanisms whereby Tregs protect against atherosclerosis are multiple (Figure 1). The main cytokines produced by Tregs, IL-10 and TGF-β, have strong anti-atherogenic activities in murine models. Gene deletion, or inhibition of IL-10 or TGF-β with neutralizing antibodies, aggravates atherosclerosis in mice and exacerbates effector Th1 and Th2 responses [77,78,79]. However, given that both TGF-β and IL-10 are not exclusively produced by Tregs, but can also be released by macrophages, these studies did not provide evidence of causality between Tregs and protection against atherosclerosis. Yet, overexpression of IL-10 by T cells reduced atherosclerosis in *Ldlr*^−/−^ mice [80], and inactivation of TGF-β signaling specifically in T cells exaggerated atherosclerosis [81,82], providing evidence for an important role of these cytokines in Treg-dependent atheroprotection. 

#### 4.4.2. Modulation of Immune Cell Functions

Tregs prevent T cell polarization into Th1 and Th17 subtypes, and limit their pathogenic activities. They can also modulate macrophage functions. They inhibit the proinflammatory properties of macrophages and shift macrophage differentiation toward an anti-inflammatory phenotype [83,84]. Foam cell formation is reduced when macrophages are co-cultured with Tregs, and CD36 and SRA expression are significantly down-regulated [83]. Tregs also enhance macrophage efferocytosis by secreting IL-13 and subsequent IL-10 production in macrophages [85]. Endothelial cell activation and leukocyte recruitment can also be regulated by Tregs, independent of their immunosuppressive activities on T cells [62]. Monocyte recruitment into atherosclerotic lesions is impeded through the inhibition of MCP1 expression in DCs and macrophages [86].

#### 4.4.3. Tolerogenic DCs

DCs are the most potent APCs, with the unique property of inducing the maturation and differentiation of naïve CD4^+^ T cells by antigen processing and MHC II presentation, expressing membrane-bound costimulatory molecules, and secreting cytokines [87]. Immature tolerogenic DCs, expressing low levels of costimulatory molecules, promote tolerance by inducing T cell death and apoptosis, or deviation of immune response [88]. Activated or mature DCs can also change to a tolerogenic phenotype in response to anti-inflammatory signals, such as TGF-β or IL-10. Chemokines expressed by DCs have an important effect on the recruitment and development of Tregs: DCs expressing CCL-17 limit Treg expansion and promote atherosclerosis [89]. Inhibitory co-receptors also play important roles in the regulation of immune responses. Hematopoietic deficiency in ICOS or PD-1—inhibitory receptors of the TNF superfamily expressed by T cells—reduces Treg suppressive functions and survival and enhances atherosclerosis [31,55]. The crucial role of DCs in the regulation of adaptive immune responses has also been shown by vaccination strategies against atherosclerosis using DCs. Intravenous administration of DCs pulsed with oxLDL in *Ldlr*^−/−^ mice attenuated atherosclerosis and stabilized plaque phenotype [90]. Also, injection of DCs pulsed with ApoB-100-derived peptides and IL-10 to induce a tolerogenic phenotype in human ApoB-100-transgenic *Ldlr*^−/−^ mice has led to diminished atherosclerosis, as well as attenuated systemic and local inflammatory responses [91]. 

Indoleamine 2,3-dioxygenase (IDO), a rate-limiting enzyme involved in the catabolism of tryptophan, can promote peripheral immune tolerance by inhibiting T cell activation and proliferation [92]. IDO-expressing mature DCs can stimulate proliferation of Foxp3^+^ Tregs [93]. Conversely, CTLA-4-expressing Tregs have been shown to activate IDO in DCs, which contributes to the maintenance of DC tolerogenic phenotype [94]. Interestingly, injection of DCs rendered tolerogenic by TGFβ2 treatment into hyperlipidemic *Ldlr*^−/−^ mice promoted the expansion of FoxP3^+^ Tregs, and was associated increased IDO expression and reduced atherosclerosis [95].

#### 4.4.4. Metabolism

Finally, a large body of evidence indicates that the intracellular metabolism of Tregs controls their suppressive function, stability, and expansion capacity. Further studies are required to better understand how intracellular metabolism might be involved in the atheroprotective effects of Tregs [67].

## 5. Strategies to Promote Atheroprotective T Cell Immunity

A prior study by our group in the late 90s showing that the administration of ovalbumin-specific Tr1 cells, together with their cognate antigen, reduced atherosclerosis in *Apoe*^−/−^ mice was the first demonstration that manipulation of Tregs is a promising anti-atherosclerotic strategy. Subsequently, other strategies were developed in experimental models of atherosclerosis, based on antigen-specific or -non-specific approaches to promote Treg activation, expansion, survival, or suppressive functions, and protect against atherosclerosis (Table 1). 

### 5.1. Antigen-Specific Induction of Tregs

The induction of antigen-specific immunological tolerance has the potential to inhibit harmful immune responses to self- or allogeneic antigens, while preserving the integrity of the remaining immune system. Different strategies of immunization with LDL, oxLDL, or ApoB-related peptides have proved beneficial in experimental models of atherosclerosis [96]. Unexpected observations of reduced atherosclerosis in hypercholesterolemic rabbits immunized with oxLDL were the first indications that it could be possible to use a “vaccine” approach to treat atherosclerosis [97,98]. Subsequent studies have identified certain peptide sequences in Apo-B-100 as major antigens responsible for the immunization-induced, pro-atherosclerotic response [99,100]. Subcutaneous administration of apoB-derived peptides in *Apoe*^−/−^ mice induces antigen-specific Treg expansion, inhibits effector T cell responses, and decreases atherosclerosis [101]. Oral treatment with oxLDL promotes the expansion of Tregs in secondary lymphoid organs and inhibits atherosclerosis in *Ldlr*^−/−^ mice [102,103]. Atherosclerosis is reduced by the adoptive transfer of tolerogenic DCs pulsed with human ApoB-100 and IL-10 in transgenic *Ldlr*^−/−^ mice expressing human ApoB [91]. Notably, depletion of Tregs prevents in *Apoe*^−/−^ mice the atheroprotective effect of immunization with ApoB-peptides [104]. Interestingly, immunization of *Ldlr*^−/−^ mice with multi-antigenic epitopes from ApoB100, hHSP60, and *Chlamydophila pneumoniae*, with alum as adjuvant, was more effective in inhibiting atherosclerosis than single- or bi-epitope vaccine, associated with a stronger specific Treg response [105]. Altogether, these preclinical studies provide evidence that an antigen-specific Treg response can be induced by immunization to protect against atherosclerosis (Table 1).

### 5.2. Non-Antigen-Specific Induction of Tregs 

#### 5.2.1. Anti-CD3

Following preclinical and clinical studies showing that treatment with anti-CD3 monoclonal antibodies can restore immune tolerance in type 1 diabetes [115], this strategy was applied in experimental atherosclerosis. Anti-CD3 antibody treatment reduced atherosclerosis in *Ldlr*^−/−^ [106] and *Apoe*^−/−^ mice [107], and enhanced the expression of Foxp3 in spleen cells [106]. Oral administration of anti-CD3 also protected against atherosclerosis, as a result of the selective expansion of Tregs expressing latent associated protein (LAP), the amino-terminal domain of the TGF-β precursor peptide [116]. In line with these findings, in vivo neutralization of TGF-β prevented the atheroprotective effects of anti-CD3 antibodies [117].

#### 5.2.2. G-CSF

The growth factor granulocyte-colony-stimulating factor (G-CSF), often used during cancer treatments to correct chemotherapy-induced neutropenia, has been shown to modulate T cell and DC functions. Interestingly, treatment with G-CSF inhibits atherosclerosis in *Apoe*^−/−^ mice, increases the number of Tregs in lymph nodes, spleen, and atherosclerotic lesions, and enhances their suppressive function in vitro [108]. 

#### 5.2.3. Vitamin D3

Based on studies showing that calcitriol, the active form of vitamin D3, is a potent immunomodulator, its effect was evaluated in atherosclerosis. Oral treatment with calcitriol reduced atherosclerotic lesions in *Apoe*^−/−^ mice via the induction of Tregs and tolerogenic DCs [109].

#### 5.2.4. FTY720

Similarly, the sphingosine-1-phosphate receptor agonist FTY720 (Fingolimod), which increases the functional activity of Tregs, inhibits atherosclerosis in *Apoe*^−/−^ when orally administered at low doses by the expansion of Tregs, reduces effector T cell responses, and increases TGF-β expression [110]. 

#### 5.2.5. mTOR

Another way to regulate Treg differentiation and function is through mammalian target of rapamycin (mTOR) signaling. Rapamycin, an inhibitor of mTOR, has potent immunosuppressive properties, induces Treg expansion, and depletes effector T cells [118]. Recently, it has been shown in *Ldlr*^−/−^ mice that increased intracellular cholesterol in Tregs due to T cell–specific deficiency in ABCG1 leads to mTOR inhibition, which promotes Treg development and results in protection against atherosclerosis [119].

#### 5.2.6. Measles Virus

The nucleoprotein of the measles virus has been shown to block DC activation [120]. Treatment of *Apoe*^−/−^ mice with this viral nucleoprotein markedly reduces atherosclerosis and induces a Tr1-type regulatory immune response characterized by enhanced IL-10 but reduced IFN-γ and IL-4 production [111].

#### 5.2.7. Low-Dose IL-2

IL-2 is essential for Treg homeostasis in vivo [38]. IL-2 combined with the IL-2-specific monoclonal antibody (JES6-1) has been shown to specifically expand FoxP3^+^ Tregs in mice [121]. Notably, in clinical studies, low-dose IL-2 has increased circulating Treg levels in patients with chronic graft-versus-host disease, with no effect on effector T cell levels [122]. A substantial proportion of treated patients showed significant clinical improvement. In patients with autoimmune vasculitis induced by hepatitis C virus, treatment with was also able to improve clinical symptoms and increased the percentage of circulating FoxP3^+^ Tregs [123]. A similar beneficial effect of low-dose IL-2 treatment was observed in patients with type 1 diabetes [124]. In *Apoe*^−/−^ mice, local delivery of IL-2 to atherosclerotic lesions [112] or treatment with IL-2/anti-IL-2 mAb immunocomplexes [113,114] lead to a reduction in atherosclerosis due to Treg expansion. These findings in experimental atherosclerosis are being translated into clinical applications, with the first ongoing double-blind, placebo-controlled phase I/II clinical trial, LILACS, aiming to assess the safety and efficacy of low-dose IL-2 in patients with stable ischemic heart disease and in patients with ACS [125].

## 6. Conclusions

The three positive clinical trials in the last years, CANTOS [1], ColCot [2], and LoDoCo2 [3], have provided exciting evidence that targeting inflammation reduces the risk of another cardiovascular event in patients who had a prior heart attack. The anti-inflammatory agents used in these trials, anti-IL-1β and Colchicine, act on the innate immune system. Yet there is a large body of evidence accumulated from animal and human studies to indicate that adaptive immune responses also contribute to the initiation and progression of atherosclerosis, and that Tregs are crucial atheroprotective modulators. Strategies that boost the development of Tregs by using biologicals or induce their antigen-specific generation permit the specific blockade of the deleterious effects of self-reactive immune destruction while maintaining the ability of the immune system to clear non-self-antigens. Several prototype atherosclerosis vaccines have demonstrated beneficial effects in experimental animal models by inducing Treg-dependent tolerance against LDL-associated antigens. In this respect, they resemble the immunization strategies developed for the prevention of other autoimmune diseases. However, translating these strategies from the bench to the bedside for the treatment of atherosclerosis is still challenging, and represents a stimulating endeavor, which ultimately may improve cardiac outcomes in patients with CVD. 

Another exciting approach for treatment of atherosclerosis is Treg therapy, based on a phenomenon known as “infectious tolerance” [126]. The principle is that the anti-inflammatory and tolerogenic effects of Tregs promote the differentiation of recipient T cells into new Tregs, even after the transferred Tregs are eliminated. Adoptive cell therapy with Tregs is a promising therapeutic strategy to treat autoimmune diseases, because sustained tolerance and suppression of inflammation is ultimately mediated by the recipient’s own immune system [127]. In humans, Treg therapy was pioneered in hematopoietic stem cell transplantation [128,129], and has subsequently been shown to be safe in a number of transplant- and autoimmune-related diseases [130]. New phase II trials are ongoing to treat autoimmune diseases and prevent graft rejection [131]. Preclinical studies have shown that the adoptive transfer of Tr1-like Tregs reduces experimental atherosclerosis [58]. The development of similar Treg-based therapeutic strategies in humans could be a promising, attractive approach to combat atherothrombotic diseases.

## Figures and Tables

**Figure 1 cells-10-00723-f001:**
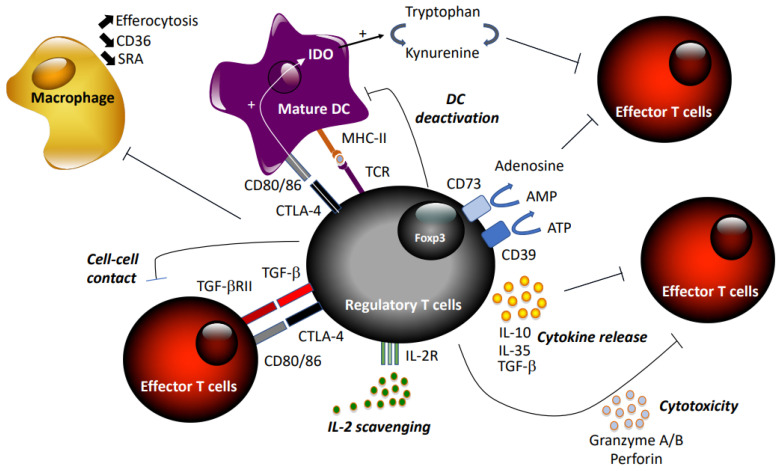
Summary of suppressive mechanisms of regulatory T cells.

**Table 1 cells-10-00723-t001:** Strategies developed in animals to expand athero-protective regulatory T cells. Apoe, Apolipoprotein e; CD: Cluster of Differentiation; DC, dendritic cells; FoxP3, Forkhead/winged helix transcription factor 3; G-CSF, Granulocyte- Colony-Stimulating Factor; HSP, Heat Shock Protein; IFN-γ: Interferon-γ; IL, InterLeukin; LAP, Latency-associated peptide; LDLr, Low Density Lipoprotein receptor; MDA-LDL, Malondialdehyde-LDL; TGF-β, Transforming growth factor- β; Th, T Helper; TNF-α, Tumor Necrosis Factor- α; Treg, regulatory T cell.

	Antigen	Route of Delivery	Species	Immune Effects	References
Antigen-specific induction	MDA-LDL Copper-oxidized LDL	Subcutaneous + Freund’s adjuvant	Rabbit	Increased auto-antibodies titers	[97,98]
	apoB-derived peptides	Subcutaneous	*Apoe*^−/−^ mice	Increased Foxp3+ Tregs Decreased Th1 and Th2 signature	[101]
	apoB-derived peptides	Subcutaneous + Freund’s adjuvant	*Apoe*^−/−^ mice	Increased Foxp3+ CCR5+Tregs Increased IL-10 production	[100]
	Ox- and MDA-LDL	Oral route	*Ldlr*^−/−^ mice	Increased Foxp3+Tregs Increased TGF-β production	[102]
	apoB-derived peptide (aBp210)	Subcutaneous + Alum	*Apoe*^−/−^ mice	Increased Foxp3+ CD25+ Tregs Decreased Th1 and Th2 signature	[104]
	ApoB100	Loaded on DC stimulated by IL-10	*HuB100*(*tg*) *X* *Ldlr*^−/−^ mice	Decreased T effector proliferation Decreased Th1 signature	[91]
	HSP 60	Oral route	*Ldlr*^−/−^ mice	Increased Foxp3+ CD25+Tregs Increased IL-10, TGF-β production	[103]
	ApoB100 peptide + HSP60 peptide + Chlamydophila pneumoniae peptide	Subcutaneous + Alum	*Apob^tm2Sgy^* *Ldlr ^tm1Her J^ mice*	Increased Foxp3+ CD4+ T cells Increased IL-10 and TGF-β production Decreased IFN-γ and TNF-α production	[105]
Non-antigen specific induction	Anti-CD3 alone or combined with IL-2	Intravenous	*Ldlr*^−/−^ mice	Increased Foxp3+Tregs Decreased IFN-γ and TNF-α production	[106]
	Anti-CD3	Oral route	*Apoe*^−/−^ mice	Increased Foxp3+Tregs Decreased Th1 and Th2 signature Increased TGF-β production	[107]
	G-CSF	Subcutaneous	*Apoe*^−/−^ mice	Increased CD4+ CD25+Tregs	[108]
	vitamin D3	Oral route	*Apoe*^−/−^ mice	Increased Foxp3+Tregs Decreased CD80+CD86+ DC Increased IL-10 production	[109]
	FTY720	Oral route	*Apoe*^−/−^ mice	Increased LAP+Foxp3+ cells Decreased Th1 signature Increased TGF-β production	[110]
	Nucleoprotein of measles virus	Intraperitoneal	*Apoe*^−/−^ mice	Decreased T effector proliferation Decreased Th1 signature Increased IL-10 production	[111]
	Low IL-2	Intravenous	*Apoe*^−/−^ mice	Increased Foxp3+ CD25+ Tregs	[112]
		Intraperitoneal	*Ldlr*^−/−^ mice	Increased Foxp3+ CD25+ Tregs Decreased Th1 and Th2 signature	[113]
		Intraperitoneal	*Apoe*^−/−^ mice	Increased Foxp3+ CD25+ Tregs Decreased Th1, Th2, Th17 signature	[114]

## Data Availability

Not applicable.

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
