# Peer review of "Regulatory T Cell-Enhancing Therapies to Treat Atherosclerosis"

_cells, 2021, doi:10.3390/cells10040723_

Round 1

Reviewer 1 Report

Ait-Oufella et al. presents an interesting review on the role of Treg cells in protection of atherosclerotic lesions, pointing out the Treg based therapeutic approach. The review however would benefit from more deep analysis of CD8+ T cell role in atherosclerosis. The Authors should refer to recent review on this topic, published btw. in Cells by Schafer and Zernecke. In this context, the role of CD8+ Treg cells should also be mentioned, as a subset of Treg cells. In the section 4.2 and Figure 1, IDO (indoleamine-2,3 dioxygenase) should be introduced as one of the most crucial mechanism of Treg activities.  

Author Response

Response: we would like to thank the reviewer for his/her encouraging comments. As suggested, several paragraphs regarding the role of CD8 T (page 4), regulatory CD8 T cells (page 8)and and IDO (page 11) have been inserted in the manuscript and in the figure 1.

Reviewer 2 Report

In the presented manuscript “Regulatory T Cell Enhancing Therapies to Treat Atherosclerosis” written by Ait-Oufella et. al., the authors summarized the information about the role of T cells in atherosclerosis.  Manuscript is well written, but does not contain any new/ unique perspective, which would distinguish manuscript compared to other already published reviews.

Author Response

Response: we would like to thank the reviewer for his/her comments. As suggested, at the end of the manuscript a paragraph has been inserted with new perspective. We introduced the concept of Treg-mediated “infectious tolerance » that can be translated to cardiovascular diseases (Page 14).